# Relationship between Postpartum Metabolic Status and Subclinical Endometritis in Dairy Cattle

**DOI:** 10.3390/ani12030242

**Published:** 2022-01-20

**Authors:** Uxía Yáñez, Pedro G. Herradón, Juan J. Becerra, Ana I. Peña, Luis A. Quintela

**Affiliations:** Unit of Reproduction and Obstetrics, Department of Animal Pathology, Faculty of Veterinary Medicine, Universidade de Santiago de Compostela, Avda. Carballo Calero s/n, 27002 Lugo, Spain; uxia.yanez.ramil@usc.es (U.Y.); garcia.herradon@usc.es (P.G.H.); juanjose.becerra@usc.es (J.J.B.); ana.pena@usc.es (A.I.P.)

**Keywords:** uterine infection, calving, neutrophils, immune response, negative energy balance, ketosis

## Abstract

**Simple Summary:**

Proper reproductive efficiency is decisive to achieving adequate profitability in dairy farms. However, uterine pathologies such as subclinical endometritis (SE) play a primary role in the decline of reproductive performance. This disease impairs reproductive function, and its incidence may reach up to 34% during the first seven weeks after calving. Consequently, identifying the predisposing factors and diagnosing this pathology as early as possible is mandatory to minimize the impact on the profitability of the farms. Several metabolic alterations postpartum have been related to the occurrence of SE, so our objective was to identify which alterations act as risk factors for SE. Uterine and blood samples and data from 94 Holstein cows were collected 30–45 days after calving. Our results showed that serum levels of β-hydroxybutyrate acid (BHBA), albumin, and urea are related to the incidence of SE, being BHBA a predisposing factor and albumin and urea protective factors. Therefore, these metabolites should be carefully considered during the postpartum period as indicators of SE. Additionally, preventive measures aimed to control these alterations may be useful to prevent SE.

**Abstract:**

The aim of this study was to verify the importance of postpartum serum levels of certain metabolic markers as risk factors for subclinical endometritis (SE). Ninety-four Holstein cows were included in the study, and examinations were carried out between 30–45 days postpartum. Rectal palpation, vaginoscopy, transrectal ultrasound, endometrial cytology, and blood sample collections were performed. The percentage of polymorphonuclear neutrophils (%PMN) on the endometrium was evaluated, as well as serum levels of glucose, cholesterol, triglyceride, albumin, hepatic enzymes, urea, non-esterified fatty acids (NEFA), and β-hydroxybutyrate acid (BHBA). Samples with ≥8% PMN were classified as positive to subclinical endometritis. According to the serum levels of BHBA, cows were classified as clinical ketosis (>2.6 mmol/L), subclinical ketosis (1.2–2.6 mmol/L), and healthy (<1.2 mmol/L). Additionally, body condition score, parity, date of last labor, peripartum issues, insemination date, date of pregnancy diagnosis and milk production information were collected. Data were analyzed using a multiple regression analysis. The results showed that as serum levels of BHBA rose, also did the %PMN, so that up to 60% of cows with clinical ketosis suffered from SE. On the other hand, the %PMN fell as serum levels of urea and albumin increased. Consequently, good postpartum management practices and early detection of metabolic alterations are necessary measures to control predisposing factors and reduce the incidence of SE.

## 1. Introduction

Milk production and reproductive efficiency are the main determining factors of dairy farms profitability [1]. In recent decades, we have witnessed a gradual decline in the reproductive performance of these farms [2,3,4], although some researchers have recently claimed that several fertility traits had shown improvement [5]. Moreover, it was stated that reproductive disorders, including infertility [6], are one of the main causes of cow culling [7].

With no doubt whatsoever, uterine pathology plays a primary role in the decline of reproductive efficiency. Metritis and endometritis are related to delays in postpartum return of ovarian activity, increasing calving-first service and calving-conception intervals, decreasing pregnancy rates, increasing number of services per conception, and higher culling rates [8,9,10,11,12]. Since the beginning of this century, new diagnostic techniques have been implemented to improve the detection of these uterine diseases, such as ultrasonography [13,14] and uterine cytology [13,15], being the last one especially useful for diagnosing subclinical endometritis.

Clinical endometritis (CE) can be defined as the inflammatory process of the endometrial lining of the uterus accompanied by a purulent or mucopurulent vaginal discharge, in absence of systemic signs of illness [16,17]. On the other hand, subclinical endometritis (SE) do not present clinical signs of illness and are characterized by the presence of polymorphonuclear neutrophils (PMN) on the endometrium, and none or little exudate inside the uterus [18,19]. It should be noted that both issues occur ≥21 days after calving.

The prevalence of CE varies from 9.6% [20] to 28% [21] and 67.2% [22]. However, when it comes to SE, it is difficult to compare the prevalence among studies, as it depends on the day of diagnosis after calving, the technique used to perform the endometrial cytology (cytobrush or uterine infusion), as well as the cut-off point of the percentage of PMN (%PMN), and the sample. Nevertheless, SE should be carefully considered, as its prevalence may reach up to 34% during the first seven weeks after calving [13,19].

It is noteworthy that there is high uterine contamination during the days following calving (until 10–14 days postpartum in most cows), being the most common microorganisms: *E. coli, Streptococcus, T. pyogenes, B. licheniformis, Prevotella spp.,* and *F. necrophorum* [17]. The relaxation of the physical barriers that normally prevent the entrance of microorganisms into the uterus, and the existence of a suitable culture media inside it (blood, remaining necrotic tissue) are responsible for this occurrence. When these microorganisms adhere to the mucosa, colonize, or penetrate the epithelium and/or release bacterial products (toxins, enzymes), it can be considered a uterine disease [17].

In this regard, the main factor that prevents the occurrence of uterine diseases after calving is the innate immune function. It has been proven that during this period, there is a noticeable expression of genes in endometrial cells that codify different elements related to the innate immune system activity: Type Toll receptors, mediators of inflammation, and effector molecules [23,24]. Furthermore, the arrival of large amounts of PMN also takes place. In healthy animals, this innate immune function is able to eliminate bacterial contamination. Therefore, the development of clinical signs depends on the balance between the microorganisms that colonize the uterus, the immune response of the animal, and the environment [25]. Any factor that may disrupt this balance would favor the occurrence of postpartum uterine diseases.

At the beginning of lactation, dairy cows are not able to meet the high energy needs due to their low dry matter intake after calving, which can be reduced to 50% in some animals. This causes a situation of negative energy balance (NEB). During the NEB period, fat mobilization takes place and increases the supply of non-esterified fatty acids (NEFA) and the serum levels of β-hydroxybutyrate acid (BHBA). Both metabolites are indicators of an adaptative response to NEB and have been associated with the impairment of PMN function, cytokine production, and bacterial recognition [1,26]. Given the critical role immune function plays in uterine health during postpartum, the relation between these metabolic markers and uterine diseases should be clarified.

Considering the prevalence of SE and its negative effects on reproductive performance in cattle, the prevention of this disease is vital to maintain the profitability of the farms. Presently, there is still controversy about which postpartum metabolic alterations are related to this issue, as different results among studies can be observed [27,28,29,30]. Therefore, the aim of this study was to verify the importance of the serum levels of the abovementioned metabolites at 30–45 days postpartum as risk factors for SE.

## 2. Materials and Methods

### 2.1. Animals, Farms, and Management

This study included a total number of 94 Holstein cows (parity 1–8, mean = 2.61) and was conducted on a population of 25 dairy farms located in the Northwest of Spain, with an average of 35 cows in milk per farm (15–90). Parity season took place between October and February. The mean 305-day milk production was 8610 Kg. All farms had a conventional milking parlor, and cows were milked twice a day.

Regarding the design of the facilities, 15 farms (60%) were tie-stall farms and 10 (40%) were free-stall farms. In all farms, cows were fed a total mixed ration, and a reproductive examination was carried out every 15 to 30 days, depending on the herd size.

### 2.2. Selection of Animals

To select the animals, the reproductive status was evaluated between days 30 and 45 of lactation. Those animals in the luteal phase or anestrous were included in the study and blood samples were taken. On the contrary, those animals in estrous at the time of the examination were excluded.

Thereafter, rectal palpation, vaginoscopy, and transrectal ultrasound were performed in 94 cows. Those cows with perceptible signs of clinical endometritis (i.e., abnormal uterine fluid discharge, abnormal uterine contents, or absence of uterine involution) detected by, at least, one of the abovementioned techniques, were classified as positive to clinical endometritis. Finally, uterine cytology was carried out in all the remaining animals (*n* = 51) with no signs of endometritis.

### 2.3. Cytology Sample Collection

The endometrial cytology was carried out with a small cytobrush (length 20 mm and diameter 0.6 mm) located inside an insemination rod (Quicklock 2000, Minitube Iberica, Barcelona, Spain). Both the rod and the cytobrush were protected by an insemination sheath and covered with a plastic sanitary sleeve (Chemise Sanitaire, IMV Technologies, L’Aigle, France).

The rod, covered with the insemination sheath and the plastic sanitary sleeve, was introduced through the vagina to the external cervical opening. Inside the cervix, the sanitary sleeve was perforated by the insemination rod that was passed into the uterine body. There, the plunger was pushed to externalize the cytobrush and a sample was collected by rotating the cytobrush against the adjoining uterine wall. Before its complete removal, the cytobrush was replaced inside the rod to prevent cellular contamination from the cervix or vagina.

After the cytobrush was removed, the sample was extended on a glass slide via rotation and was allowed to air-dry. Slides were stained with Diff Quick (Quick Panoptic kit, Quimica Clinica Aplicada S.A., Tarragona, Spain). Each slide was examined by the same person, using an optical microscope at 100x magnification (CHT, Olympus Iberia S.A.U., Barcelona, Spain). To perform a quantitative assessment of endometrial inflammation, at least 150 cells (excluding erythrocytes) were counted. Samples with ≥8% PMN were classified as positive to subclinical endometritis [31], whereas samples with <8% PMN were classified as healthy.

### 2.4. Blood Sample Collection

After the genital examination, blood samples were taken by puncture of the coccygeal vein using vacuum tubes without anticoagulants. Serum samples were separated after centrifugation at 1000 g for 20 min and stored in 0.75 mL aliquots at −20 °C until analysis.

All analyses were performed using a digital photometer (Selecta MD200, Barcelona, Spain), except total protein analysis, for which a portable refractometer was used (J.P. SELECTA S.A., Abrera, Barcelona, Spain). Glucose, total cholesterol, triglyceride (TAG), and albumin concentration were determined by a colorimetric endpoint method (Biosystems S.A., Barcelona, Spain). Hepatic enzymes alanine aminotransferase (ALAT), and aspartate aminotransferase (ASAT) were determined by IFCC (International Federation of Clinical Chemistry) using Biosystems reagents. Urea was also analyzed by a colorimetric enzymatic method with Spinreact reagents (Spinreact, S.A.U., Sant Esteve de Bas, Spain). Non-esterified fatty acids (NEFA) and β-hydroxybutyrate acid (BHBA) were determined by kinetic enzymatic kits (Randox Laboratories Ltd., Antrim, United Kingdom). According to the serum levels of BHBA, cows were classified into three categories: clinical ketosis (>2.6 mmol/L, *n* = 5), subclinical ketosis (1.2–2.6 mmol/L, *n* = 6)), and healthy (<1.2 mmol/L, *n* = 40) [32].

### 2.5. Data Collection

Specific data from each animal were provided by a collaborator veterinarian, who collected all the information on the software ReproGTV: body condition score (BCS), assessed at the time of sample collection; parity, date of the last labor, insemination date, date of pregnancy diagnosis, and milk production information. Furthermore, the presence of the following peripartum issues was also collected: labor assistance (human intervention), retention of fetal membranes (presence of the fetal membranes in the genital tract 24 h after calving), abortion (fetal loss >45 days after artificial insemination), hypocalcemia (depressed, laying-down cow that recovers after treatment with calcium), ketosis (presence of ketone bodies in urine, decreased milk production and food intake), and mastitis (macroscopic alterations of milk).

### 2.6. Statistical Analysis

First, the obtained data were analyzed using a Pearson correlation to preselect the variables that were most related to the %PMN, selecting those parameters with *p* ≤ 0.1 for the following analysis. Next, a multiple linear regression analysis was carried out. The %PMN was chosen as the dependent variable, and the factors selected in the previous analysis were used as independent variables. Normality was checked by the Kolmogorov–Smirnov test (*p* > 0.05). The model met the assessments of linear regression. All analyses were conducted in SPSS version 20.0 for Windows (SPSS INC, Chicago, IL, USA). Differences were considered significant at *p* ≤ 0.05.

## 3. Results

The results for the correlation showed that BCS, urea, BHBA, NEFA, and calving-sample collection interval were significantly related to the percentage of PMN (*p* ≤ 0.05). These parameters, along with albumin (*p* ≤ 0.1), were included in the multiple linear regression analysis (Table 1).

The results for the multiple linear regression analysis (Table 2) showed that urea, albumin, and BHBA were significantly related to the percentage of PMN (*p* ≤ 0.05). The effect of BHBA was remarkably higher (B coefficient = 5.39) than the effect of the remaining variables (B coefficient = −0.23 and −0.58 for urea and albumin, respectively).

As serum levels of urea (Figure 1) and albumin (Figure 2) rose, the % PMN fell. Regarding BHBA, the opposite effect can be observed: as the serum levels of BHBA soared, so did the % PMN (Figure 3).

As is depicted in Figure 4, 60% of the animals with clinical ketosis also suffered from subclinical endometritis. This number decreased to 13.5% when healthy cows were considered.

## 4. Discussion

Our results showed that the occurrence of SE is significantly related to the serum levels of albumin, BHBA, and urea 30–45 days postpartum.

The prevalence of SE is favored by metabolic imbalances, such NEB [29]. During the first two weeks postpartum, cows manifest NEB, and it may be extended due to nutritional, environmental, or management issues. Overall, serum levels of NEFA and BHBA, as well as a decreasing BCS, are considered as indicators for NEB [33]. Due to NEB, cows mobilize their bodily reserves, releasing NEFA into the blood circulation, which will be metabolized by the liver, turning into ketone bodies or kept as triglycerides, which may lead to fatty liver disease [26]. The last week before parturition, the recommended serum levels of NEFA are <0.5 mMol/L [34]. Galvao et al. [35] observed that cows with SE had greater plasma NEFA concentrations at 35 DIM than healthy cows. In this regard, Pascottini and LeBlanc [30] also found greater serum concentrations of NEFA 15 days postpartum for cows with SE compared to healthy cows. It was proven that high serum levels of NEFA induce an increase in Reactive Oxygen Species (ROS), which reduce the viability of neutrophils and lead to an impairment of the immune system function [36]. Although we did not find a significant relation between serum levels of NEFA and the occurrence of SE, according to the initial correlation this factor was included in the regression. In fact, it was observed that cows with higher serum NEFA concentrations tended to suffer from SE (*p* = 0.115). In the present study, samples were obtained between days 30 and 45 after calving, which may explain the lack of significance between this variable and %PMN, as NEFA values remain constant as of the first month postpartum [37].

Simultaneously, there is an increase in serum levels of ketone bodies that leads to the well-known ketosis, with its functional and productive consequences for the animal. Among these ketone bodies, BHBA is the more stable one. Therefore, the measure of BHBA level is the gold standard for the diagnosis of ketosis [38]. Our findings showed that as serum levels of BHBA rose, also did the %PMN, so that up to 60% of cows with clinical ketosis (BHBA > 2.4 mmol/L) suffered from SE (%PMN > 8). Therefore, our results support the statement that ketosis favors the occurrence of uterine infections. This is in accordance with findings from other researchers, who also observed that increased serum levels of BHBA were related to the occurrence of SE [1,27,30,33,35]. One possible explanation is that high levels of BHBA impair the function of PMN, reducing chemotaxis and phagocytosis, consequently leading to immunosuppression [39]. Nevertheless, no association between ketosis and SE was found by Shin et al. [29]. Although there were differences among studies regarding the type of housing, management, and the number of animals included, further research is needed to shed light on this controversial issue.

On the other hand, our results revealed that both albumin and urea may have a positive effect on the uterine condition. These findings agree with those recently obtained by Pascottini and LeBlanc [30], who observed greater albumin concentrations in healthy cows than in cows with SE 35 days postpartum. In addition, Burke et al. [40] also stated that cows with low %PMN had greater albumin concentrations. Additionally, low serum levels of albumin may indicate impaired liver function and have been related to high serum levels of BHBA and NEFA [41]. Moreover, they were also associated with low, sustained protein levels, independent of the dietary protein content [41]. During NEB, the triglycerides stored in fat tissue suffer lipolysis, giving rise to glycerol and NEFA. The latter pass into the bloodstream non-covalently bound to serum albumin. As a consequence, modifications on albumin levels might lead to unbound NEFA and their storage in the liver as triglycerides, which may cause fatty liver and aggravate the NEB [42,43]. Furthermore, Schneider et al. [44] found that, considering its role as a negative acute-phase protein, serum albumin concentrations may be of aid to identify, before calving, cows at risk of developing uterine diseases, since its levels were significantly reduced 3 weeks prepartum in cows diagnosed later with uterine infection.

Little is known about the role urea plays in fertility, as contradictory results among studies were obtained. In our study, we observed that high serum levels of urea were related to a decrease in %PMN and, therefore, to a decrease in the probabilities of with SE. Although further research is needed to establish the relation between urea and SE, this effect may be associated with pH modifications (i.e., acidification) caused by this metabolite. pH modifications are one of the various defense mechanisms of the female genital tract, the same as alterations in the composition of genital secretions, antibody titer and characteristics, alterations in the fermentative ability of microorganisms of the uterine flora, and changes in the composition of the reticuloendothelial system [45].

On the other hand, our results show no significant relation between the occurrence of SE and glucose, in accordance with those obtained in a recent study [30]. This may be explained by the fact that cows have low blood glucose, and its concentrations are tightly controlled by homeostasis, making glucose an inadequate metabolite for monitoring herd problems [46]. However, it was stated that plasma glucose concentration is associated with the occurrence of CE [47]. Additionally, in this study, no relation was found between SE and cholesterol and total protein levels. On the contrary, Pascottini and LeBlanc [30] observed lower serum cholesterol at 14, 21, and 35 days postpartum, and greater total protein concentrations at 14 days postpartum, in cows with SE. Due to differences in methodology among studies, further research is required to clarify if these factors are associated with the risk of SE.

## 5. Conclusions

This study shows that there is a relation between serum levels of BHBA, albumin, and urea 30–45 days postpartum and the increase in %PMN and SE. Considering the negative effects of this issue on reproductive performance and the profitability of the farm, it is necessary to implement measures to control these predisposing factors and reduce the incidence of this disease. Good postpartum management practices, a proper BCS at calving, and avoiding factors that worsen the NEB are preventive actions that may improve the immune response and diminish the incidence of SE.

It is also noteworthy that further research is needed to verify if the application of new technologies in-farm, such as BHBA-meters in milking robots, may be of aid to perform early detection of metabolic alterations in order to refine and correct herd management mistakes that might lead to postpartum reproductive pathologies.

## Figures and Tables

**Figure 1 animals-12-00242-f001:**
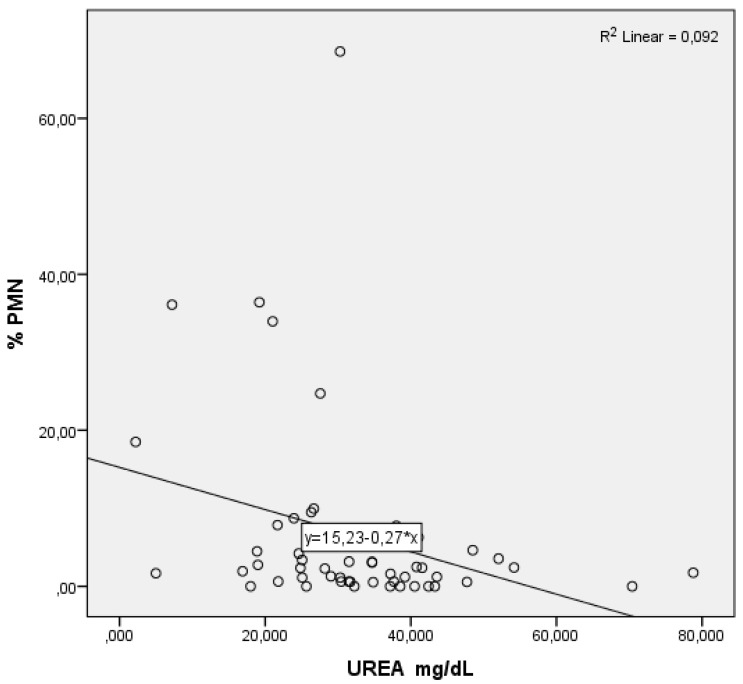
Partial regression graphic of urea (independent variable) and its relation with the percentage of polymorphonuclear neutrophils (%PMN, dependent variable).

**Figure 2 animals-12-00242-f002:**
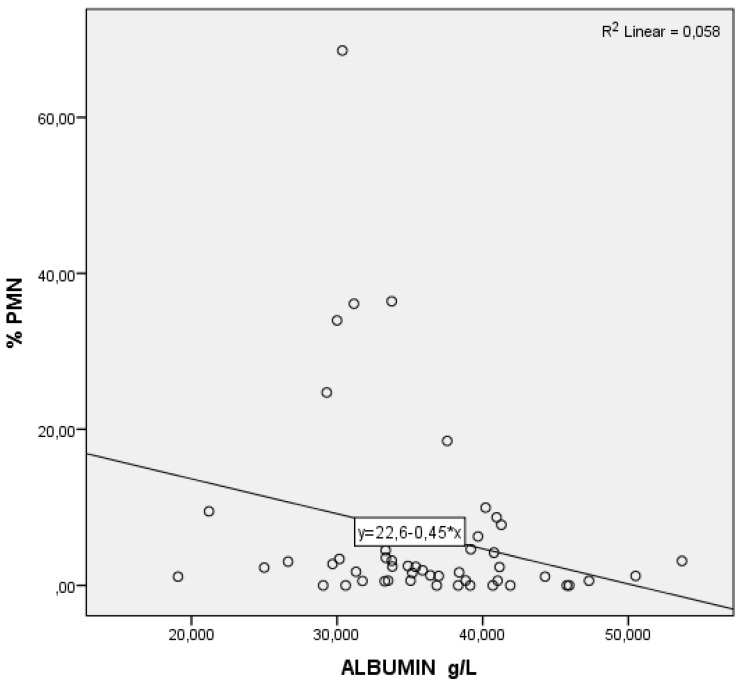
Partial regression graphic of albumin (independent variable) and its relation with the percentage of polymorphonuclear neutrophils (%PMN, dependent variable).

**Figure 3 animals-12-00242-f003:**
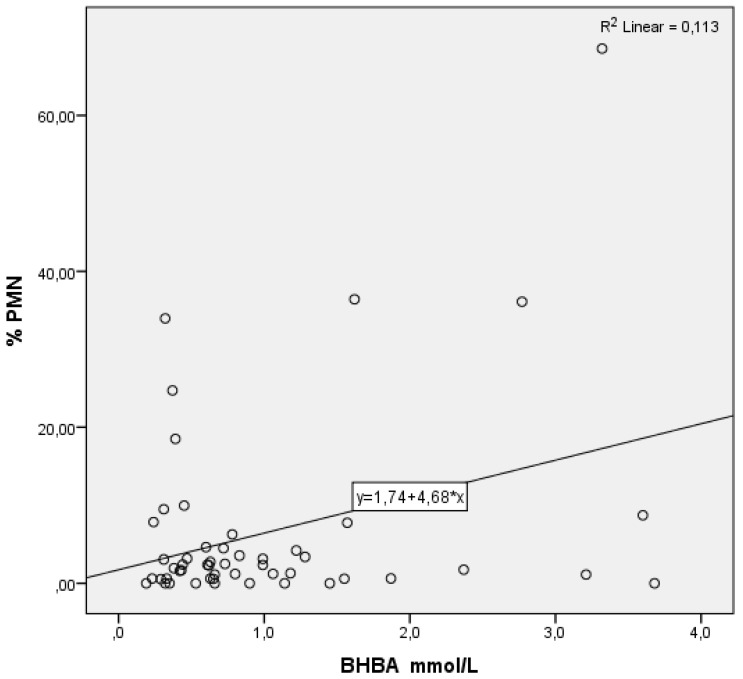
Partial regression graphic of β-hydroxybutyrate acid (BHBA, independent variable) and its relation with the percentage of polymorphonuclear neutrophils (%PMN, dependent variable).

**Figure 4 animals-12-00242-f004:**
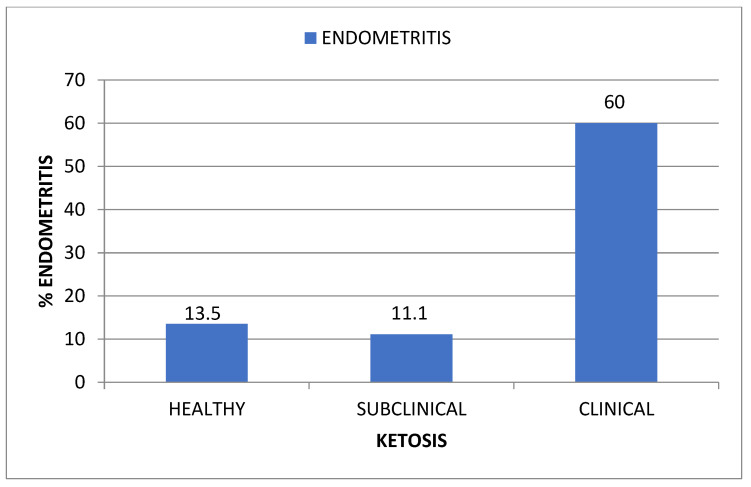
Incidence of endometritis (% endometritis) regarding the occurrence and the type of ketosis.

**Table 1 animals-12-00242-t001:** Results of Pearson correlation between the analyzed variables and the percentage of polymorphonuclear neutrophils (%PMN).

	Correlation with %PMN
305-d milk production	−0.10
Glucose	−0.24
Cholesterol	−0.14
Triglyceride	0.03
Albumin	−0.24 ^†^
Total protein	−0.08
Urea	−0.30 *
BHBA	0.34 *
NEFA	0.34 *
Calving-sample collection interval	−0.37 **

BHBA: β-hydroxybutyrate acid; NEFA: non-esterified fatty acids; ^†^
*p* ≤ 0.1; * *p* ≤ 0.05; ** *p* ≤ 0.01.

**Table 2 animals-12-00242-t002:** Resulting model for the multiple linear regression analysis. The percentage of polymorphonuclear neutrophils was selected as the dependent variable and the remaining factors as independent variables.

Model	B Coefficient ± Standard Error	Significance
Constant	63.19 ± 18.18	0.001
BCS	−5.86 ± 4.37	0.189
Calving-sample collection interval	−0.46 ± 0.30	0.129
Urea (mg/dL) *	−0.23 ± 0.10	0.026
Albumin (g/L) *	−0.58 ± 0.23	0.014
NEFA (mmol/L)	3.13 ± 1.94	0.115
BHBA (mmol/L) **	5.39 ± 1.92	0.008

BCS: Body Condition Score; BHBA: β-hydroxybutyrate acid; NEFA: non-esterified fatty acids; * *p* ≤ 0.05; ** *p* ≤ 0.01.

## Data Availability

Data can be provided by the correspondence author.

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
