# Peer review of "Relationship between Postpartum Metabolic Status and Subclinical Endometritis in Dairy Cattle"

_animals, 2022, doi:10.3390/ani12030242_

Round 1

Reviewer 1 Report

The paper entitled “Relationship between postpartum metabolic status and subclinical endometritis in dairy cattle” evaluated the association between serum metabolites and the incidence of subclinical endometritis. The results showed that BHBA is positively correlated with the % of PMN in the endometrium, while albumin and urea are negatively associated with % of PMN. The study is interesting; however, the introduction needs to be improved in order to show how this study answers the gaps of studies already published in the literature. There are important studies that were not included, and there are studies that did similar evaluations and are only cited in the discussion.

See comments below:

Abstract

Line 22: Please change “94” for “Ninety-four”.

Line 25-27: Please specify the type of sample, since PMN and metabolites were not evaluated in the same tissue.

Introduction

Line 42: Although for a long time, dairy farms have experienced a decline in reproductive performance recently, the reproductive outcomes are stable or slightly increasing. Two of the references used are from almost 20 y ago. Please check more recent references, as for example Norman et al. (2019), available in: https://queries.uscdcb.com/publish/dhi/current/reproall.html

Line 44: Change “elimination” for “culling”

Line 67: Please change “delivery” for “calving”

Line 71: Please change the sentence “contamination does not imply disease”

Line 77: Expression of genes in which cell?

Line 81: “normal state of sterility”. The uterus is not a sterile environment. Studies already detected the presence of bacteria before calving, and also after calving in cows that did not develop any uterine disease. Please check the review from Galvao et al. (2019) https://doi.org/10.3168/jds.2019-17106

Line 85-93: The critical period in which NEFA and BHB are increased is in the first three weeks postpartum. Please add the reasoning why the measurement of BHB between 30-45 d is important.

Line 96: Why controversy? Please add references that support your statement.

Line 98: The authors used the word “modifications” in the objective. However, it was only collected one blood sample per cow, in this sense, the study does not provide an idea of the dynamics of metabolites, only an observation in a specific time-point. In this sense, I suggest changing the objective according to the design.

The topic of this study is relevant, however, it is missing the citation of important literature in the area, as for example Dubuc et al. (2010) https://doi.org/10.3168/jds.2010-3429, who already described the risk factors for endometritis. The authors should include this citation in the introduction and point the gaps in the knowledge and how the current study was designed to fill up those gaps. In the study from Dubuc et al. (2010), blood sampling was performed early postpartum, while the current study performed the sampling after one month. Why do the authors believe this time point is better?

Methodology

Line 120: Please explain why animals in estrus were excluded

Line 145: Please change “g” for “ g”

Line 160: When BCS was evaluated?

Line 162: Define how hypocalcemia, ketosis and mastitis were diagnosed.

Line 165-171: Please add all the variables used for the correlation analysis. Why the authors did not include the effect of the farm?

Did cows with clinical endometritis were excluded from the analysis?

Results

Figure 3: The authors are using “BHB” as the legend of the axis when over the whole test is used “BHBA”. I suggest keeping the acronyms constant.

Figure 4: Please centralize the number label of each column. The number for subclinical ketosis is overlaid with the graph design, which I suggest changing. How many cows were detected with clinical/subclinical ketosis?

Just as a suggestion: the authors could perform logistic regression with the incidence of endometritis according to the incidence of ketosis.

Discussion

Line 210: Please define “issues”

Line 211: Please put the second and third paragraphs together

Line 254-266: The discussion of the urea result is explaining about a possible effect in the endometrium. However, the authors measured urea in serum and not in the uterus. Also, the authors did not measure uterine pH. Please do not discuss possible outcomes that were not evaluated in the current study.

Line 268-269: Please change the sentence. NEFA was significantly correlated with %PMN, glucose was not.

Reviewer 2 Report

The study aimed at investigating the possible relation between the metabolic status of postpartum cows at 35 DIM and the subclinical endometritis

The paper is really well written, the study well-conducted and the results clearly presented together with a nice discussion

I have only few minor comments regarding study design:

  • I don't see (my fault maybe) any results/comments in the discussion regarding the relationship between calving events (dystocia/abortion etc) and subclinical endometritis. Is there a greater incidence in dystocic calvings or abortion or twinning or retained placenta?
  • possibly add some discussion on changes of albumin considering its role as a negative acute-phase protein 
  • it would have been interesting to use blood samples to detect also possible changes in circulating PMN and total WBC. Maybe think about adding this to a next phase study
